# Nanoscale Structural Phase Transitions in Aqueous Solutions of Organic Molecules

Nikolai Bunkin [1] , Leonard Sabirov [2] , Denis Semenov [2],*, Faxriddin Ismailov [2] and Muxriddin Khasanov [2]

1 Department of Fundamental Sciences, Bauman Moscow State Technical University, Moscow 105005, Russia; nbunkin@mail.ru
2 Institute of Engineering Physics, Samarkand State University, Samarkand 140104, Uzbekistan
* Correspondence: denis.samarkand@gmail.com

**Abstract:** Adiabatic compressibility $\beta_S$ of the 4-methylpyridine + water solution is investigated in a wide concentration and temperature variation interval using Mandelstam–Brillouin scattering spectroscopy. The adiabatic compressibility minimum caused by the microinhomogeneous structure of the solution is experimentally established at the concentration of 0.06 molar fractions of 4-methylpyridine in the solution. The results of the investigations allow the construction of a diagram of possible states caused by a continuous three-dimensional hydrogen bond network of water. The results of experimental study of the excessive hypersound absorption in acetone + water and 3-methylpyridine + water solutions are discussed based on the conclusions of the theory of high-frequency sound scattering near the critical point (developed by Chaban) and the Landau theory. These results are described within the framework of the Landau and Chaban theories and explained by the existence of two different states with minimum thermodynamic stability in the solution.

**Keywords:** light scattering; hypersound; compressibility; sound velocity; absorption coefficient; aqueous solution





## 1. Introduction

In addition to binary solutions that have the upper or lower critical stratification point, there is a small group of solutions which, while being homogenous over the entire plane of the state diagram in coordinates $x$, $t$ ($x$ is the concentration of one of the components, $t$ is the temperature), exhibit nontrivial behavior of some physical parameters on approaching a "critical" concentration $x_0$ and temperature $t_0$ (e.g., maxima in temperature and concentration dependences of integrated light scattering intensity [1] and specific heat capacity [2], minimum of the diffusion coefficient [3], maximum of the Landau–Placzek ratio in the Rayleigh triplet of the scattered light [4]).

It is assumed that the solution state with the coordinates $x_0$ and $t_0$ (so-called singular point) is most close to stratification, and the observed phenomena are due to the increase in the concentration fluctuations, as is the case near critical stratification points. This assumption is confirmed, for example, by the fact that a closed stratification region appears in these solutions when a small amount of a third component is added [5] or pressure is varied [6].

Investigation of hyperacoustic parameters of the single point of the guaiacol–glycerin solution [7] showed that behavior of the hypersound velocity and absorption coefficient experimentally observed in the vicinity of $t_0$ was not typical of ordinary liquids, which indicated manifestation of physical mechanisms that were not directly related to critical concentration fluctuations but rather were caused by the difference of the internal solution structure on both sides of $t_0$.

The relation between the appearance of a singular point in the solution and the possibility of existence of different structural phases in liquids (with the guaiacol–glycerin

solution taken as an example), which was substantiated in [7], is physically important and requires experimental confirmation with other binary systems.

One of the parameters closely related to the structure of a liquid is compressibility. Experimental values necessary for the calculation of adiabatic compressibility by the Laplace equation ($\beta_S = 1/\rho V^2$, where $V$ is the speed of sound, and $\rho$ is the density), are easy to measure with a high accuracy, which is why determination of this and its derivative characteristics (excessive molar adiabatic compressibility, partial adiabatic compressibility [8]) is a popular method for studying the solution structure.

Note that the literature data on adiabatic compressibility of solutions are obtained from measurements of the speed of ultrasound. Data on elastic properties of solutions at higher frequencies (hypersonic) are hardly available. At the same time, it is shown in [7,9,10] that in the vicinity of the solution single point in the temperature dependencies of the hypersound velocity and absorption coefficient there are some features never observed in ultrasonic experiments.

In [11–13], we investigated the 4-methylpyridine (4MP) + water system in a wide range of solution temperature and concentration variation using Mandelstam–Brillouin scattering spectroscopy. It was established that regularities of frequency-shift kinetics of Mandelstam–Brillouin components (MBCs) could be due to restructurings in the solution that arises from both a change in its temperature and a change in the nonelectrolyte concentration. Lines of "critical" points (temperature $t^*$ and concentration $x^*$) were found, which separate solutions with different structural organizations of components. Based on these results, we proposed a qualitative diagram of structural states of solutions that reflected boundaries of existence of a continuous hydrogen bond network in solutions. However, regularities of concentration-related compressibility variation in the vicinity of the singular point were not investigated.

Thus, light scattering spectroscopy is an effective method for studying critical phenomena in liquids, which allows one to gain a much deeper insight into the nature of the critical state. The studies of the indicatrix of scattered light intensity near the critical point, as well as the measurements of the spectral width of light scattered by order-parameter fluctuations, confirmed the scaling theory predictions for the behavior of some equilibrium and kinetic coefficients near the critical point [14]. However, despite the successes of the scaling theory, far from all specific features of the critical state of matter have been studied. First of all, the character of high-frequency sound propagation near the so-called "singular point" (minimum thermodynamic stability) of solutions has been poorly investigated experimentally. Research in this field is necessary for developing deep-seated oil and gas deposits, in which organics are in the supercritical states, because high-frequency acoustic spectroscopy provides missing information about the correlation properties of liquids at short distances.

The experimental and theoretical study of the processes occurring near the critical point is an urgent line of research in modern physics because the theory of condensed matter has been insufficiently developed. At the same time, the phenomena in the range of second-order phase transitions occur very similarly; i.e., the critical phenomena exhibit isomorphism. Due to this important property of critical phenomena, one can choose a convenient object of study and then transfer the obtained results to other (unstudied) objects.

An analysis of the character of propagation of ultrasound and hypersound yields important information about the dynamics of critical phenomena. The hypersound velocity and attenuation are usually found from the Mandelstam–Brillouin scattering spectra [15]. These spectra are due to the light scattering from adiabatic density fluctuations.

The propagation of ultrasound and hypersound near critical points and second-order phase transitions has been experimentally studied for a long time. Fixman [16] and Kawasaki [17], based on the theory of interacting modes, described the behavior of the ultrasound absorption rate and absorption coefficient near critical stratification points. The formulas of these theories, as well as their further modifications, are in satisfactory agreement with the experimental data obtained in the ultrasonic frequency range, where

$\Omega\tau \leq 1$ ($\Omega$ is the frequency of sound and $\tau$ is the relaxation time of order-parameter fluctuations). In the high-frequency range, where $\Omega\tau > 1$, the formulas of the aforementioned theories predict a decrease in the critical contribution to the sound absorption coefficient. At $\Omega\tau \gg 1$, the behavior of the kinetic properties is described more adequately in terms of dynamic scaling theory [18]. In this case, the characteristic size rc (correlation radius) in the system is replaced by the sound wave number $q^{-1}$, and, according to the theory, the kinetic properties cease to depend on the proximity to the critical point.

An attempt to develop a general theory of sound propagation and light scattering in solutions with a singular line, solutions with a singular point, and in stratified solutions was made by Chaban [19] taking into account the nonlocality, which, in combination with the theory of interacting modes, provided expressions describing the experimental data at $\Omega\tau > 1$ as well. Using this theory, one can describe quite well the behavior of the dispersion of the ultrasound velocity and absorption coefficient near critical points, up to $\Omega\tau \approx 50$.

However, the attempts to apply the theory to the calculation of the attenuation coefficient of hypersound in a solution with an immiscibility dome were unsuccessful [20], which stimulated the search for new approaches to the description of the hyperacoustic properties of solutions in the vicinity of the stratification point, the double critical point, and the singular point corresponding to the thermodynamically unstable state.

This work reports the results of our experimental studies of hypersonic parameters of aqueous 4MP, 3MP, and acetone solutions to reveal regularities of processes related to variation in the structural organization of the solution components as the singular point is approached both in temperature and in concentration. The results we present in this paper were partly discussed in [21,22].

## 2. Results and Discussion

### 2.1. 4MP-Water Solution: Adiabatic Compressibility in a Solvent-Rich Region

Figure 1 shows isotherms of $\beta_S$ dependences on solution concentrations calculated from the data on MBCs frequency shifts. The form of the dependence $\beta_S(x)$ appreciably changes with varying temperature. In the low-temperature region, $\beta_S$ nonmonotonically changes with decreasing 4MP concentration in the solution and passes through the minimum at a certain concentration.

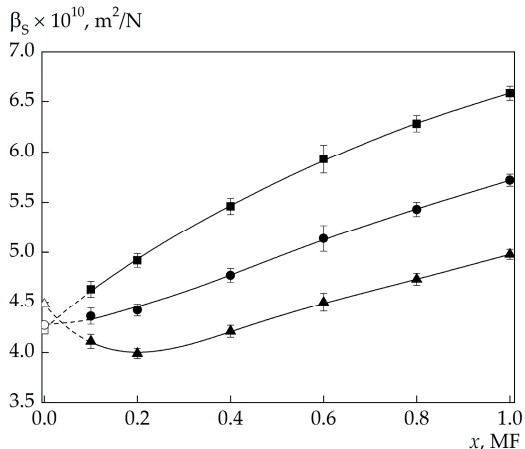

**Figure 1.** Concentration dependences of $\beta_S$ in 4MP–water solutions at the temperatures of 25 (▲), 45 (●), and 65 °C (■). Dashed lines are the extrapolation of the form of the dependence to the value at $x = 0$ (pure water), $\Delta$, $\bigcirc$, and $\square$ are the values of $\beta_S$ for pure water at these temperatures.

As the temperature increases, the position of the minimum in the dependence $\beta_S(x)$ shifts to the region of lower concentrations, and the minimum itself becomes less distinct. Finally, in the high-temperature region, the minimum in the $\beta_S(x)$ isotherms disappears, and $\beta_S$ is seen to monotonically decrease with decreasing concentration.

The position of the $\beta_S(x)$ minimum on the concentration scale at different temperatures, $x_{\min}(t)$, is well described by the expression

$$x_{\min} = A\exp\left[-\frac{t}{B}\right] + C,\qquad(1)$$

where $A$ = –0.00467, $B$ = –12.21285, and $C$ = 0.25412; the units of measurement of $x_{\min}$ and $t$ are molar fractions (MF) and the degree Celsius, respectively.

The presence of the minimum in the $\beta_S(x)$ isotherms is also confirmed by the results of the investigations of the concentration dependence of the 4-MHz ultrasound velocity [23]. Analysis of adiabatic compressibility isotherms shows that the minimum in the $\beta_S(x)$ isotherms for aqueous 4MP solutions is observed in the region of medium concentrations in both the ultrasonic [23] and the hypersonic experiment. Moreover, the concentration at which $\beta_S(x)$ passes through the minimum does not change in going from ultrasonic to hypersonic frequencies. In the investigated solutions, the transition from the ultrasonic frequency range (4 MHz) [23] to hypersonic frequencies leads to a change in the sound wavelength from $4 \times 10^{-4}$ to $\approx 5 \times 10^{-7}$ m.

The fact that inversion of the concentration dependence of adiabatic compressibility is identically well "sensed" by ultra- and hypersonic waves indicates the presence of structural changes in water at certain concentrations of 4MP in solutions both on a small (comparable to the hypersound wavelength) and a large (comparable to the ultrasound wavelength) spatial scale, i.e., the changes actually make a global effect on the entire system.

### 2.2. 4MP-Water Solution: Adiabatic Compressibility in a Water-Rich Region

Figure 2 shows the temperature dependences of $\beta_S$ for low-concentration solutions calculated from the data on the MBCs frequency shift at the scattering angle of 90°. Additionally shown is the temperature dependence of $\beta_S$ for pure water. It is seen that $\beta_S$ variation with increasing solution temperature is nonmonotonic. As the temperature increases, $\beta_S$ first decreases, passes through the minimum at a certain temperature, and then increases with further increasing temperature.

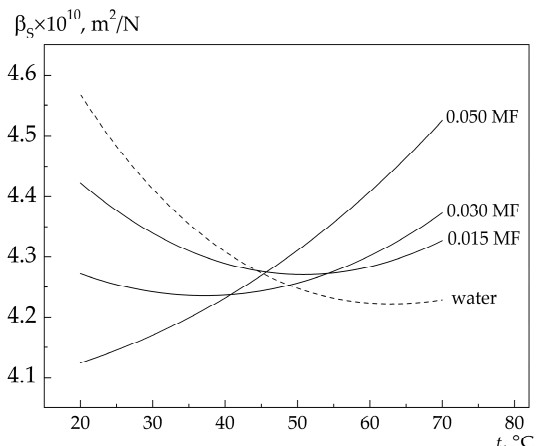

**Figure 2.** Temperature dependences of adiabatic compressibility measured at the hypersonic frequency for low-concentration aqueous 4MP solutions. The hypersonic frequency is $\approx$4.6 GHz.

The position of the minimum on the temperature scale depends on the concentration of 4MP in the solution. For example, the minimum of adiabatic compressibility in pure water is observed at $\approx$63 °C. In solutions with the concentrations of 0.015 and 0.03 MF, the minimum is at $\approx$51 and $\approx$37 °C, respectively.

The results presented in Figure 2 show that in 4MP–water solutions of low concentrations the derivative of $\beta_S(t)$ with respect to t undergoes sign inversion at a certain

temperature (at fixed solution concertation). The inversion point (temperature) depends on the solution concentration and shifts to the region of lower $t$ with increasing $x$.

The concentration dependences of $\beta_S$ at low concentrations and temperatures 25 and 65 °C are shown in Figures 3 and 4. Apart from $\beta_S$ for the investigated solutions, $\beta_S$ for solutions of 4MP with the concentrations of 0.1 and 0.2 MF are also given for illustrative purposes. The results of the experiment reveal an additional minimum in $\beta_S(x)$ appearing at low solution concentrations. The concentration width of this minimum strongly depends on temperature, while its depth hardly depends on temperature and is about 4% of the "background" compressibility.

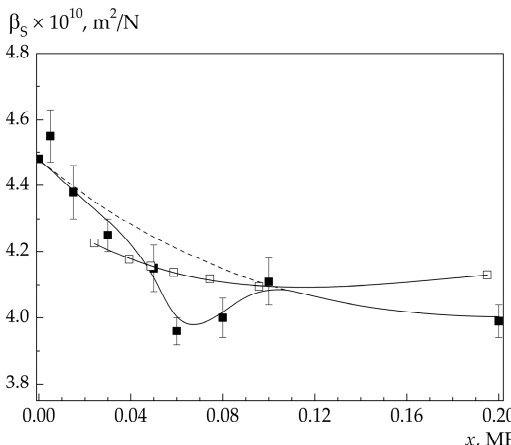

**Figure 3.** Dependence of adiabatic compressibility on the 4MP concentration in the solution at 25 °C. The sound frequency is ≈4.7 GHz (■) and 4 MHz (□) [23].

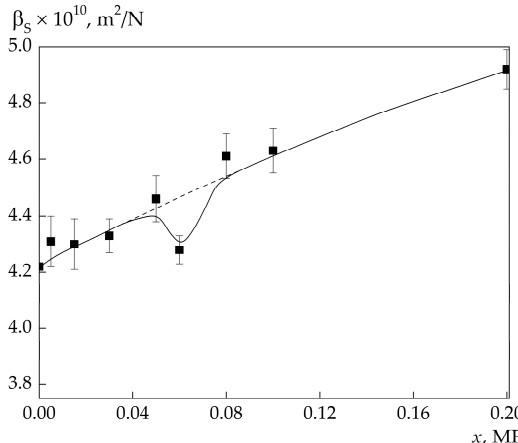

**Figure 4.** Dependence of adiabatic compressibility on the 4MP concentration in the solution at 65 °C. The sound frequency is ≈4.7 GHz.

"Background" compressibilities for solutions with concentrations $x < 0.1$ MF shown in Figures 3 and 4 by the dashed line were determined by extrapolation of $\beta_S(x)$ at $x \geq 0.1$ MF into the region of lower concentrations to the $\beta_S$ for water ($x = 0$), as shown in Figure 1.

Since in water there is no dispersion of the speed of sound in the frequency range of $10^6$ to $10^{10}$ Hz ($dV/df = 0$) [15], values of $\beta_S$ for water at hypersonic frequencies coincide with the calculations from temperature dependences of density and ultrasound propagation velocity.

It is seen in Figure 3 that the greatest difference between the "background" and experimental compressibilities, $\Delta\beta_S = \beta_S^0 - \beta_S$, is in the solutions with the concentrations of 0.06 and 0.08 MF. As the temperature in the solution with the concentration of 0.08 MF increases, $\Delta\beta_S$ decreases, and at $t = 65$ °C we have $\Delta\beta_S \approx 0$. In the solution with the concentration of

0.06 MF, in the entire investigated temperature interval, and $\Delta\beta_S$ is noticeably larger than the experimental errors.

According to the experimental results [23], variation in the ultrasound velocity ($f$ = 4 MHz) with variation in the concentration of the 4MP–water solutions in the region $x \leq 0.1$ MF is monotonic. Consequently, no additional minimum is observed at $x \leq 0.1$ MF in the concentration dependencies of $\beta_S$ measured in [23].

*2.3. 4MP-Water Solution: Structural States Diagram*

The main features of the temperature–concentration behavior of adiabatic compressibility in aqueous 4MP solutions in the hypersonic frequency range can be summed up as follows.

1.  In the $\beta_S(x)$ isotherms, a minimum is experimentally observed in the region of medium concentrations (see Figure 1); its position on the concentration scale does not depend on the sound frequency but depends on the solution temperature. As the temperature increases, the minimum shifts to lower concentrations. In other words, sign inversion of the derivative of $\beta_S(x)$ with respect to $x$ (at $t$ = const) and the inversion point shift to the region of lower $x$ with increasing $t$ are experimentally observed.
2.  In the region of low concentrations, sign inversion of the temperature coefficient $\beta_S$ (derivative of $\beta_S(t)$ with respect to $t$ at $x$ = const) occurs, and the inversion point shifts to the region of lower temperatures with increasing $x$ (see Figure 2).
3.  In the $\beta_S(x)$ isotherms, an additional minimum is found in the region of low concentrations (Figures 3 and 4). Its position on the concentration scale does not depend on the solution temperature. For ultrasound, no $\beta_S$ minimum in the region of low concentrations is experimentally observed in our investigated solutions [23].

The regularities revealed in the temperature–concentration behavior of compressibility of 4MP–water solutions confirm our earlier assumption [11] that scattered light spectra contain information on processes occurring in solutions at both the global level (in the entire system as a whole) and the local level on the scales of about tens of intermolecular distances.

Figure 5 shows lines of sign inversion points of the derivatives of adiabatic compressibility with respect to the temperature and concentration. According to (1), extrapolation of the character of the compressibility minimum shift to the region of higher temperatures (see Figure 5) predicts presence of some "critical" temperature $t \approx 48\,^{\circ}\mathrm{C}$ in pure water ($x$ = 0 MF). This temperature agrees with the results [24] obtained from the comparative analysis of temperature dependences of sheer viscosity of water and liquid argon, which allowed a conclusion that the hydrogen bond network in water suffers discontinuity at temperatures above 47 °C. Consequently, the line of sign inversion points of the derivative $d\beta_S(x)/dx$ (i.e., of such $x$ and $t$ at which $d\beta_S(x)/dx = 0$ in the investigated solutions) indicates the boundary of existence (in temperature–concentration coordinates) of the continuous hydrogen bond network in pure water and in aqueous 4MP solutions of different concentrations.

Figure 5 also shows the line of sign inversion of the derivative $d\beta_S(t)/dt$ (i.e., of such $x$ and $t$ at which $d\beta_S(t)/dt = 0$ in the investigated solutions).

The analysis of the behavior of $\beta_S$ in a wide interval of temperatures and concentrations allows the four regions (phases) with different states of solutions to be separated in the plane of the temperature–concentration coordinates.

In phase I (Ph I), which occupies a relatively narrow concentration interval, an increase in the solution temperature (at fixed concentration) or concentration (at fixed temperature) leads to a monotonic decrease in $\beta_S$. In other words, in phase I the derivatives of $\beta_S$ with respect to the temperature ($d\beta_S/dt$) and concentration ($d\beta_S/dx$) are negative.

In phase II (Ph II), an increase in $t$ (at $x$ = const) leads to a monotonic increase in $\beta_S$, while an increase in $x$ (at $t$ = const) is still accompanied by a decrease in $\beta_S$, i.e., $d\beta_S/dt > 0$ and $d\beta_S/dx < 0$.

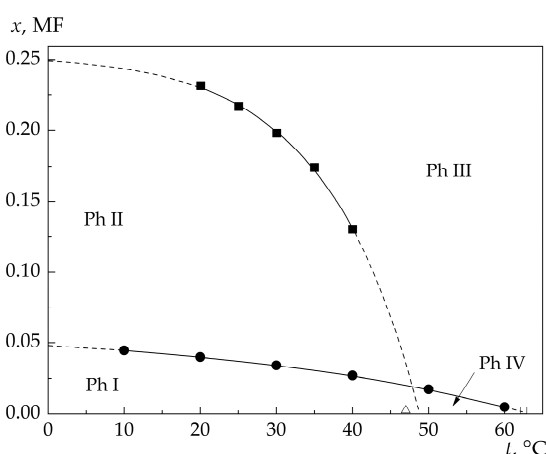

**Figure 5.** State diagram (relative to $\beta_S$ variation regularities) of aqueous 4MP solutions in the temperature–concentration coordinates (■ is the coordinates of the sign change of the derivative $d\beta_S/dx$, ● is the coordinates of the sign change of the derivative $d\beta_S/dt$): Ph I: $d\beta_S/dx < 0$, $d\beta_S/dt < 0$; Ph II: $d\beta_S/dx < 0$, $d\beta_S/dt > 0$; Ph III: $d\beta_S/dx > 0$, $d\beta_S/dt > 0$; Ph IV: $d\beta_S/dx > 0$, $d\beta_S/dt < 0$.

In phase III (Ph III), an increase in both $t$ (at $x = \text{const}$) and $x$ (at $t = \text{const}$) is accompanied by a monotonic increase in $\beta_S$, i.e., $d\beta_S/dt > 0$ and $d\beta_S/dx > 0$.

Apart from the above three phases, which confirm interpretation of the results of studying the shift of the Mandelstam–Brillouin components in the scattered light spectra [11], in 4MP–water solutions there is the fourth phase (Ph IV), in which solution compressibility increases with increasing concentration at the fixed temperature ($d\beta_S/dx > 0$), while an increase in the solution temperature (at fixed concentration) is followed by a decrease in the solution compressibility ($d\beta_S/dt < 0$).

Independence of these effects from the sound frequency (wavelength), as well as the presence and coordinates of sign inversion, points of the derivatives of compressibility with respect to temperature and concentration indicate a change in the state of solutions on the scales globally characterizing the entire system. Structural changes are determined by the role that the intermolecular hydrogen bonding plays in processes responsible for the structural state of the solution components.

It is known that compressibility of pure water decreases with increasing temperature. This behavior of compressibility probably arises from that at low temperatures the hydrogen bond network of water is not much distorted as compared to the tetrahedral configuration, and when temperature changes, restructuring of this network is of primary importance, determining the anomalous contribution to the behavior of compressibility [25,26]. At high temperatures, when the water network is strongly deformed (and perhaps partially fragmented), its restructuring affects compressibility to a lesser extent, and water behaves like all ordinary liquids.

In the region of low concentrations and temperatures (Ph I in Figure 5), nonelectrolyte molecules penetrate into the matrix of the H-bond network formed by water molecules without distorting it.

The structure of the solution in this region of temperatures and concentrations is generally identical to the structure of pure water. A decrease in compressibility of solutions with increasing concentration indicates a stabilizing (strengthening) effect of nonelectrolyte impurity molecules on the structure of the solution caused by (i) penetration of nonelectrolyte molecules into accessible hollows of the lacy structure of the hydrogen bond network without distorting its tetrahedral configuration and (ii) ejection of nonelectrolyte molecules by the water H-bond network to the places of thermal defects of the network. Actually, the decisive role is played in this region of temperatures and concentrations by hydrophobic interaction between water and nonelectrolyte molecules.

As the concentration of nonelectrolyte molecules increases, the H-bond network grows deformed but retains its three-dimensional integrity. In this region of temperatures and concentrations (Ph II in Figure 5), stabilization of the H-bond network (decrease in compressibility) continues, but now it is due to the intermolecular interaction priority change from hydrophobic to hydrophilic. Under competition for formation of H-bonds, the ability of 4MP molecules to participate in this formation with water molecules grows in importance.

With further increase in the concentration of nonelectrolyte molecules, the continuous hydrogen bond network in the solution suffers destruction (fragmentation), which leads to greater and greater increase in the solution compressibility with increasing nonelectrolyte concentration (Ph III in Figure 5). The degree of network fragmentation becomes higher with increasing concentration and temperature of the solution.

As the temperature increases, the degree of thermal deformation of the network increases, and consequently destruction of its integrity begins at lower nonelectrolyte concentrations. For this reason, the "critical" concentration responsible for the solution compressibility minimum shifts to the region of lower concentrations. It is this shift that affects the temperature–concentration behavior of the adiabatic compressibility minimum (see Figure 5).

At the same time, one more phase (phase IV in Figure 5) is observed in the region of low concentrations; in this phase, there is no continuous H-bond network in the solution, but the structure of the solution locally retains the properties characteristic of the structure of pure water (decrease in compressibility with increasing temperature). In this region of temperatures and concentrations, the number of nonelectrolyte molecules is great (at fixed temperature) for preserving the continuous H-bond network in the solution as a whole; nevertheless, their presence produces a local stabilizing effect on the network fragments. This relatively small temperature–concentration interval should probably be considered as a region of the "clustered" state of solutions if the cluster is taken to mean a spatial region of water molecule ordering, which preserves the structure of pure water due to the stabilizing effect of nonelectrolyte impurity molecules.

A transition from one phase to another can be through either a change in the solution temperature (at fixed concentration) or a change in the solution concentration (at fixed temperature).

### 2.4. 4MP-Water Solution: Nano-Scale Inhomogeneity in the Vicinity of the Singular Point

The temperature–concentration behavior of compressibility in low-concentration solutions turns out to depend on sound frequency (wavelength). The additional compressibility minimum in $\beta_S(x)$ isotherms, whose position and depth are almost independent of temperature, is only observed in the hypersonic frequency range and is not found in ultrasonic frequency experiments (Figure 3). Consequently, the additional compressibility minimum indicates local "strengthening" of the solution structure on scales comparable to the hypersound wavelength or lower, i.e., it indicates the microinhomogeneous structure of low-concentration solutions. Ultrasound is insensitive to inhomogeneity of the solution structure because of a relatively large wavelength, and thus the medium is continuous for it. That is why no additional compressibility minima are observed in solutions of low concentrations in the region of ultrasonic frequencies.

It is worth noting that the compressibility minimum is at the 4MP concentration of 0.06 MF in the solution. According to the accepted views (see, for example, [1] and references therein), the state of the 4MP–water solution with the concentration of 0.06 MF at ≈70 °C is closest to stratification (so-called singular point) and is characterized by a high level of concentration fluctuations in the vicinity of the singular point temperature. However, independence of the minimum position and depth from closeness to the singular point temperature indicates that deviation of the adiabatic compressibility $\beta_S$ from its "background" value in the solution with the concentration of 0.06 MF observed in our investigations is not directly related to the temperature region of the thermodynamic stability

minimum existing in this solution. Thus, this deviation in our investigated solutions is caused by physical mechanisms that are not directly related to the closeness of the solution to the stratification state.

When sound propagates in a spatially inhomogeneous medium, the speed of sound involves a term depending on the wave vector of the sound wave $q$ [15]

$$V^2 = V_0^2 + bq^2, \qquad (2)$$

where $V_0$ is the speed of sound in a medium without spatial inhomogeneity, and $b \approx (V_0 r)^2$ ($r$ is the characteristic scale of the spatial inhomogeneity). The result of the form (2) was obtained in theoretical studies of Vladimirskii [27] and Ginzburg [28] and, strictly speaking, is valid only at a small spatial inhomogeneity ($qr << 1$) [15].

Assuming that in the solution with the concentration of 0.06 MF there exists only one characteristic microinhomogeneity scale $r$, it can be estimated by Formula (2). Considering that $\beta_S = 1/(\rho V^2)$ and $q = 2\pi/\Lambda = \omega/V$ ($\Lambda$ is the hypersound wavelength, $\omega = 2\pi f$ is the cyclic sound frequency), Formula (2) can be transformed by simple manipulation to

$$r \approx \frac{1}{\omega \beta_S} \sqrt{\frac{\Delta \beta_S}{\rho}}. \qquad (3)$$

In Equation (3), as previously, $\beta_S$ is the adiabatic compressibility at the hypersonic frequency $\omega$, and $\Delta \beta_S$ is the difference between the "background" compressibility and $\beta_S$ at the same frequency.

Calculation by Equation (3) yields $r \approx 120$ Å for the solution with the concentration of 0.06 MF, which only slightly varies with temperature in the interval of 20 to 65 °C.

### 2.5. Acetone-Water Solution: High-Frequency Sound Velocity in the Vicinity of the Singular Point

Figure 6 shows the temperature dependencies of the velocity of ultrasound (frequency 30.3 MHz) and hypersound (frequency ~2.6 GHz) obtained by us in an acetone-water solution in the vicinity of the singular point temperature.

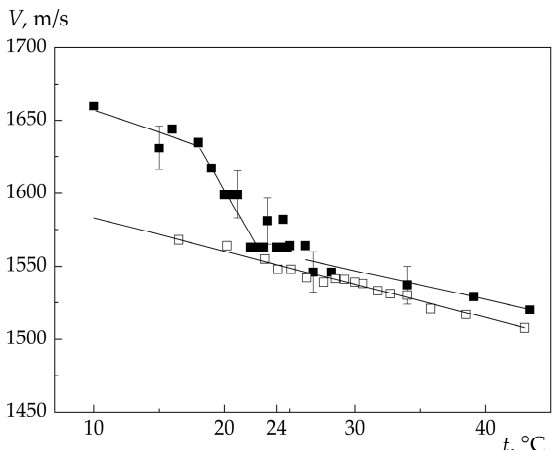

**Figure 6.** Temperature dependence of the velocity of ultrasound (□) and hypersound (■) in acetone-water solution. Singular point temperature $t_0 \approx 24$ °C.

A comparison of the experimental data obtained by us for the acetone-water solution and the authors of [7] for the guaiacol-glycerin solution shows that in the studied solutions, the temperature dependences of the high-frequency sound velocity reveal a number of patterns inherent in both solutions. At the same time, some features of the temperature dependence $V(t)$ observed in the acetone-water solution are absent in the guaiacol-glycerin solution.

Let us first list the regularities of the temperature behavior of the speed of high-frequency sound that take place both in guaiacol-glycerin and in acetone-water solutions: (1) at temperatures above the singular point, the speeds of ultrasound and hypersound depend linearly on temperature. There is practically no dispersion—the values of the velocities of ultrasound and hypersound coincide within the limits of experimental errors. The temperature coefficients ($dV/dt$) of the velocities of ultrasound and hypersound are the same; (2) in a small temperature interval near the singular point, the hypersound velocity does not depend on temperature ($dV/dt = 0$); (3) at temperatures below the singular point, the hypersound velocity also depends linearly on temperature, but with a different temperature coefficient. For ultrasound, no change in the temperature coefficient of velocity is observed.

The experimentally observed difference in the temperature coefficients of the hypersonic velocity indicates that the same solution on both sides of the singular point is described by different equations of state. Since no chemical reactions occur in the solution, it must be assumed that at temperatures above and below the singular point in a homogeneous solution, significantly different intermolecular interactions arise, leading to a change in the internal (local) structure of the solution. Ultrasound (wavelength ~$10^{-5}$ m) does not detect the influence of such structures, and for ultrasound the medium is continuous. With the propagation of hypersonic waves, the situation is different. The hypersonic wavelength (~$10^{-7}$ m) appears to be comparable with the size of the emerging structures, and therefore their difference in different temperature regions affects the value of $dV/dt$ above and below the singular point.

The experimental data obtained by us for the acetone-water solution indicate that in it, as in the guaiacol-glycerin solution, with a change (decrease) in temperature, a transition occurs from a gas-like (disordered, in the terminology of the authors of [7]) phase into the Frenkel phase (also in the terminology of the authors of [7]), characterized by the existence of an average order (ordered regions, clusters). The temperature of this phase transition coincides with the temperature of the singular point.

At the same time, in the acetone-water solution, the temperature dependence of the hypersound velocity exhibits a feature that is absent in the guaiacol-glycerin solution. Namely, below the singular point (i.e., in the Frenkel phase), one more change in the temperature coefficient of the hypersound velocity at $t \approx 18$ °C is experimentally observed (Figure 6). Thus, in contrast to the guaiacol-glycerin solution, the temperature dependence of the hypersonic velocity in the acetone-water solution below the singular point temperature is described by not one but two temperature intervals, in which the hypersonic velocity varies linearly with temperature, but with significantly different temperature coefficients. The temperature coefficient of hypersonic velocity at $t \leq 18$ °C is approximately equal to the temperature coefficient of ultrasonic velocity.

The change in the temperature coefficient of the hypersonic velocity in the temperature interval below the singular point, which was found in our experiment, is, as far as we know, the first experimental confirmation of the assumption that not one, but several Frenkel phases differing in the nature of the average order can exist in liquids.

*2.6. Acetone-Water and 3MP-Water Solutions: Hypersonic Absorption in the Vicinity of the Singular Point*

The hypersound absorption coefficients α in an acetone–water and 3-methylpyridine-water solutions, found from the MBCs spectral width, are shown in Figures 7 and 8. One can see that α changes nonmonotonically with a change in the solution temperature, and two maxima are pronounced in the dependence α(t). The position of the high-temperature maximum coincides with the singular point temperature. The temperature dependence of the excess (critical) part of absorption near the singular point temperature resembles a λ curve, which contradicts the classical theories of sound absorption near the critical point. For hypersound, the product of the frequency by the fluctuation relaxation time greatly exceeds unity ($\Omega\tau \gg 1$), and the theories predict a decrease in the critical contribution to

the absorption, whereas in our experiments we observed, in contrast, an increase in this contribution.

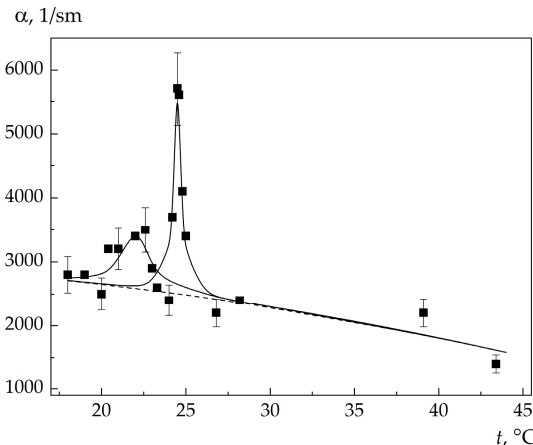

**Figure 7.** Temperature dependence of the absorption coefficient of hypersound in an acetone–water solution: solid lines are the results of calculation from the formulas of the Chaban theory [9], the dotted line shows the background (noncritical) part of the absorption.

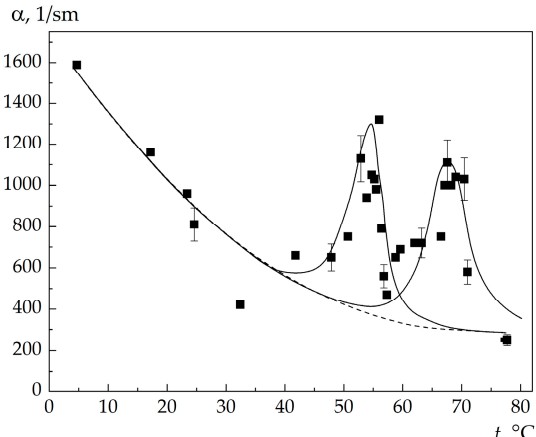

**Figure 8.** Temperature dependence of the absorption coefficient of hypersound in 3MP–water solution: solid lines are the results of calculation from the formulas of the Chaban theory [9], the dotted line shows the background (noncritical) part of the absorption.

A rise in the hypersound absorption coefficient near the stratification critical point, double critical point, and singular point has been observed experimentally previously, although only in few studies [7,10,29–32].

The inability of classical theories to describe the behavior of the absorption coefficient near the critical point stimulated Chaban to develop a modern theory [9].

Classical theories describe specifically ultrasound absorption, i.e., the loss of sound wave energy due to its conversion into heat. This conversion is caused by the delay in the change in density of the medium relative to the change in pressure in the wave when relaxation exists in the medium. However, the sound wave attenuation can be caused by both absorption of sound by inhomogeneities and scattering from them.

The Chaban theory suggests that the main mechanism of hypersound attenuation in a medium with developed fluctuations at large $\Omega\tau$ values is the scattering of a hypersonic wave from concentration fluctuations. The attenuation coefficient caused by the hypersonic wave scattering describes the Mandelshtam-Brillouin components in the scattered light spectrum, similar to the sound absorption coefficient. At ultrasonic frequencies, this contribution to the absorption is insignificant, because the ultrasound wavelength is much

larger than the fluctuation correlation length in standard experiments. However, in the case of Mandelshtam-Brillouin scattering, the hypersonic wavelength becomes comparable with the increasing fluctuation correlation length while approaching the critical point. This leads to strong scattering of hypersound.

According to Chaban [9], a change in concentration leads to a change in the density of liquid and its compressibility β. Correspondingly, fluctuations in the concentration near the critical point generate not only optical but also acoustic inhomogeneities in the medium.

Without going into the details of the theory, we present its main result, which can be used for comparison with experimental data. The Chaban theory gives the following expression for the critical part of attenuation:

$$\alpha = \frac{G}{B_1^{PP}}\left[(T - T_{PP})^2 + a_1(x - x_{PP})^2 + a_2\right]^{-\gamma}. \tag{4}$$

Here, $\gamma$ is the critical index of generalized susceptibility; $x$ is the concentration; $T_{PP}$ and $x_{PP}$ are the absolute temperature and concentration of the singular point, respectively; $a_1$, $a_2$, and $B_1^{PP}$ are constants; and $G$ is a value weakly dependent of temperature.

On the assumption that the concentration of our solution exactly corresponds to the concentration of the singular point ($x - x_{PP} = 0$), we have

$$\alpha = \frac{G}{B_1^{PP}}\left[(T - T_{PP})^2 + a_2\right]^{-\gamma}. \tag{5}$$

At the singular point temperature $T = T_{PP}$ the critical contribution to the absorption takes the maximum value $\alpha_{max} = (G/B_1^{PP})a_2^{-\gamma}$ or $G/B_1^{PP} = \alpha_{max} a_2^{\gamma}$. Substituting this expression into (5), we arrive at

$$\alpha = \alpha_{max}\left(\frac{a_2}{(T - T_{PP})^2 + a_2}\right)^{\gamma}. \tag{6}$$

The parameter $a_2$ in the theory has the meaning of the square of generalized distance (renormalized to the variation in temperature) from the singular point to the double critical point.

The key point of the theory is the choice of the critical index of generalized susceptibility $\gamma$, which determines the temperature behavior of the critical part of absorption in the vicinity of singular point. In this study, we did not make any initial assumptions about the value of $\gamma$, and the parameters $\gamma$ and $a_2$ were determined by minimizing the sum of the mean squares of deviations of experimental data from the curve calculated from formula (6). The results obtained are presented in Table 1 and Figures 7 and 8.

**Table 1.** In the acetone–water and 3MP–water solution (high-temperature maximum of hypersound absorption) and the same parameters for the low-temperature maximum.

|  | Solution | $\gamma$ | $a_2$ |
|---|---|---|---|
| High-temperature maximum | acetone-water | $1 \pm 0.05$ | $0.07 \pm 0.01$ |
|  | 3MP-water | $1 \pm 0.11$ | $10 \pm 2$ |
| Low-temperature maximum | acetone-water | $1 \pm 0.08$ | $6.6 \pm 0.5$ |
|  | 3MP-water | $1 \pm 0.10$ | $9 \pm 2$ |

As can be seen in Table 1 and Figures 7 and 8, the theory adequately describes the growth of hypersound absorption in the vicinity of the singular point temperature. Since the critical susceptibility index $\gamma = 1$, the critical dynamics of fluctuations in the vicinity of the solution singularity is described within the framework of the Landau theory.

The presence of two maxima in hypersound absorption, the previously established singularity in the behavior of adiabatic compressibility, and the consistent description of

these phenomena within the framework of the Landau and Chaban theories indicate the existence of two different states in the solutions studied, with temperature-spaced minima of thermodynamic stability.

In the vicinity of the singular point temperature, the system is characterized by a high level of order-parameter fluctuation (concentration fluctuation for the solution singular point) due to the closest proximity to the double critical point. The unreachability of the double critical point "cuts off" the fluctuation correlation length, and the fluctuation dynamics is described in the approximation of the Landau theory. At lower temperatures the system is also thermodynamically unstable, but this instability is due to the structural phase transition. In this case, order-parameter fluctuations are fluctuations of the concentration of "holes" (i.e., structureless regions). The dynamics of the "hole" concentration fluctuations is also described by the Landau theory.

### 3. Materials and Methods

Fine-structure spectra of scattered light were recorded using an experimental setup with a double-pass Fabry–Perot interferometer. The scattering angle $\theta$ was 90°. Its setting error was below 0.2°. The dispersion region of the interferometer was 0.625 cm$^{-1}$, and the contrast of the interference pattern was $5 \times 10^4$. The radiation source was a He–Ne laser (radiation wavelength $\lambda = 632.8$ nm, power 15 mW). The measurement error for the frequency shift of fine structure components $\Delta\nu$ was below 1%. Solutions were prepared with chemicals of pure grade. Optically pure components of the solution were produced by triple distillation. Solution samples were kept in sealed cylindrical glass cells at a pressure below atmospheric pressure. A cell with the sample was placed in a thermostat that allowed temperature stabilization with an error no larger than 0.05 °C. Scattering spectra were investigated in the temperature range $t = 10$–70 °C.

The coordinates of the singular point of the 3MP and 4MP–water system (according to the literature data [1]) $x_0$ and $t_0$ were 0.06 MF and about 70 °C, respectively.

The change in the scattered light frequency in the fine structure spectrum can be written as [15]

$$\frac{|\Delta\nu|}{\nu_0} = 2n\frac{V}{c}\sin\frac{\theta}{2}, \tag{7}$$

where $\nu_0$ is the frequency of the exciting light, $n$ is the refractive index, $V$ is the speed of sound, $c$ is the speed of light in vacuum, and $\theta$ is the scattering angle.

Considering (7), we calculated $\beta_S$ from the measurements of $\Delta\nu$ by the formula

$$\beta_S = \frac{1}{\rho}\left(\frac{2n\sin^{\theta}/2}{\lambda_0\Delta\nu}\right), \tag{8}$$

where $\lambda_0 = 6328$ Å is the wavelength of the exciting light, and $\rho$ is the solution density.

In the calculation of $\beta_S$ by (8) we used the results of measuring density of aqueous 4MP solutions in the temperature interval 20–65 °C [23,33] and our measured refractive index $n$ for solutions in the same temperature interval.

In the solutions studied by us, the fine structure in the spectra of scattering at the angle $\theta = 90°$ arises from modulation of the scattered light by the hypersonic waves with the frequency $f = 4.7 \pm 0.2$ GHz.

The hypersound absorption coefficient was studied by measuring the width of MBCs in the scattering spectra of linearly polarized molecular light. The MBCs full spectral width at a half maximum intensity is determined by the sound-wave temporal damping coefficient $\delta$ [15]:

$$\delta\omega_{MB} = 2\delta. \tag{9}$$

The temporal damping coefficient $\delta$ is expressed in terms of the amplitude coefficient $\alpha$:

$$\delta = \alpha V, \tag{10}$$

where $V$ is the speed of sound.

As was found in [34], when measuring the ultrasound absorption in an acetone–water solution, the acetone concentration corresponding to the maximum ultrasound absorption and minimum solution stability amounts to 0.4 molar fractions. In this study the measurements were performed specifically for a solution of this concentration. The singular point temperature is $t_0 \approx 24$ °C [35].

### 4. Conclusions

The reported results of investigating the adiabatic compressibility of aqueous 4MP solutions at the hypersonic frequency allowed existence boundaries to be experimentally established (in temperature–concentration coordinates) for different phases characterized by different structural organization of solution components. The transition between different structures can be either through changing the solution temperature (at fixed concentration) or through changing the solution concentration (at fixed temperature).

The data on adiabatic compressibility of solutions measured at the hypersonic frequency in these investigations serve as experimental evidence, confirming existence of a continuous hydrogen bond network in pure water and aqueous solutions in a certain interval of temperatures and concentrations and its transformation from the undeformed state (at low nonelectrolyte concentrations) to the deformed state followed by destruction (fragmentation) with increasing concentration of nonelectrolyte molecules in the solution.

The presence of an additional adiabatic compressibility minimum at the singular-point concentration (0.06 MF) at the hypersonic frequency and its absence for compressibility measured at the ultrasonic frequency indicate that at the singular point there is an additional mechanism for "strengthening" of the solution structure due to the phase transition of the structural type on the scales of about 10–12 nm.

In acetone-water and 3MP-water solutions the excessive spectral broadening of the spectral fine-structure components in the vicinity of the singular point is due to the interaction of adiabatic density fluctuations with order-parameter fluctuations, which leads to an increase in the attenuation coefficient of hypersound due to its incoherent scattering from order-parameter fluctuations.

The fluctuation dynamics near the singular point temperature in the investigated aqueous solution of acetone is described by the Landau theory of second-order phase transitions with a critical index of generalized susceptibility $\gamma = 1$.

The presence of two maxima in the temperature dependence of the hypersound absorption coefficient, the singularity of the behavior of the displacement of the spectral fine-structure components, and adiabatic compressibility, as well as the consistent description of the observed phenomena within the framework of the Landau and Chaban theories are due to the existence of two different states with a minimum of thermodynamic stability in the solution.

**Author Contributions:** Conceptualization, N.B., D.S. and L.S.; methodology, D.S. and L.S.; investigation, D.S., F.I. and M.K.; data curation, N.B. and L.S.; writing—original draft preparation, D.S.; writing—review and editing, N.B. and L.S.; supervision, N.B. All authors have read and agreed to the published version of the manuscript.

**Funding:** This research received no external funding.

**Data Availability Statement:** The data supporting the findings of this study are available upon request.

**Conflicts of Interest:** The authors declare no conflict of interest.

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
