# Peer review of "Nanoscale Structural Phase Transitions in Aqueous Solutions of Organic Molecules"

_condensedmatter, doi:10.3390/condmat8030064_

Round 1

Reviewer 1 Report

This paper studies aqueous solutions of 3- and 4-methylpyridine and acetone. Ultrasound and hypersound measurements of compressibility show regions that hint at singularities. These are related to variation in the proposed structural organization of the water molecules dependent upon temperature and concentration.

The data look good and the interpretation leans heavily on previous work by these authors. I have no objection to the interpretation; it’s a reasonable scenario.

Page 7 discusses a porous network of water molecules that exists below 47C and in which non-electrolyte molecules can penetrated. Can the authors give a size for the pores and a size limit for the penetrating molecules? Would the resulting structures be similar to clathrates?

It is interesting to ask what would be observed if D2O was used instead of H2O. D2O is more “water-like” than H2O.This has been done with 3MP, see "Light Scattering and Viscosity Studies of a Ternary Mixture with a Double Critical Point," J. Chem. Phys. 83, 1835 (1985).

Minor comment: Line 143 the parameters A, B and C seems to have too many significant figures 

I recommend publication.

C. M. Sorensen

Author Response

Dear Prof. Sorensen,
Thank you very much for your kind review of our article and very interesting questions and comments on it. I hope I can at least partially answer the points you are interested in.

> Can the authors give a size for the pores and a size limit for the penetrating molecules?

It seems reasonable that more or less adequate "direct" information in such a very subtle issue as the size of pores in the structure of water is given by a computer experiment. In view of the obvious problems in modeling a large continuous structure, the vast majority of works are devoted to modeling water clusters, which, according to some authors, can be considered as a kind of structure-unit of a continuous network structure. Calculations give a size (effective radius) for such a unit radius up to 8 Å [for example, Lanza G., Water model for hydrophobic cavities: structure and energy from quantum-chemical calculations, Phys. Chem. Chem. Phys., 2023, 25, 6902]. A rather rough estimate of the effective radius of 4MP molecule (https://chemapps.stolaf.edu/jmol/jmol.php?model=Cc1ccncc1 can be used for such an estimate) is about 3 Å, that suggests a possibility of its introduction (at sufficiently low concentrations) into water structure without its significant distortion.

Although, of course, the real situation may not be so unambiguous. The idea of a plurality of such structural units is more realistic. When another substance is dissolved in water, the distribution of the probability of the appearance of individual types of structural units should change and, in addition, the appearance of local structures that are not found in pure water is, of course, very likely.

>Would the resulting structures be similar to clathrates?

In a number of works, the authors refer to possible structures in aqueous solutions of some nonelectrolyte molecules as clathrate-like or quasi-clathrate. To some extent, this can be reasonable. But, as far as we know, in true clathrates, there are no strong interactions between the guest molecule and the molecules of the host lattice surrounding it. Therefore, in the case of molecules with a pronounced tendency of hydrophilicity and hydrophobicity in an aqueous medium, the appropriateness of such a formulation, in our opinion, is a matter for further studies and discussion. 

>It is interesting to ask what would be observed if D2O was used instead of H2O. D2O is more “water-like” than H2O.This has been done with 3MP, see "Light Scattering and Viscosity Studies of a Ternary Mixture with a Double Critical Point," J. Chem. Phys. 83, 1835 (1985).

This is a very interesting question. If we completely replace H2O with D2O for 3MP and 4MP, then there are no noticeable changes in the behavior of adiabatic compressibility in the ultrasonic experiment [Marczak W., Speed of Ultrasound, Density, and Adiabatic Compressibility for 4-Methylpyridine + Heavy Water in the Temperature Range 293 K to 313 K, J. Chem. Eng. Data 1999, 44, 936] In this work, the author also mentions that for aqueous solutions at the mole fraction of methylpyridine slightly smaller than 0.05, there arises probably a clathrate-like structure similar to the solid hydrate of type II. The unit cell of that structure contains 136 hydrogen-bonded water molecules and 8 large hexacaidehedral voids in which single methylpyridine molecules may be accommodated. But we are not aware of any data on hypersonic experiments in H2O+D2O+4MP or H2O+D2O+3MP solutions, in which, as was investigated in your experiment, a closed-loop coexistence curve occurs, and can be transformed into a double critical point (merge of  upper and lower consolute points) by increasing H2O/D2O ratio.

>Minor comment: Line 143 the parameters A, B and C seems to have too many significant figures

These parameters were obtained from data processing by OriginPro software (with proper +/- errors) and used further “as is” to approximate the dependence to zero-concentration (pure water) and zero-temperature (oC). Of course, the lines in Fig.5 should be considered as effective boundaries of different structural states of the 4MP-water system.

Reviewer 2 Report

That is a very nice work using Brillouin scattering spectroscopy. The authors detected microinhomogeneous structures in the solutions, and, they were able to construct a diagram of 12 possible states due to a continuous three-dimensional hydrogen bond network of water. The main results are discussed based on very important theories (Chaban and Landau), well-known by people working in this area. In my point of view, it is read to be published.

Author Response

Dear Reviewer,
Thank you very much for such a kind and positive feedback on our article.